# The Gastric Cancer Immune Prognostic Score (GCIPS) Shows Potential in Predicting an Unfavorable Prognosis for Gastric Cancer Patients Undergoing Immune Checkpoint Inhibitor Treatment

**DOI:** 10.3390/biomedicines12030491

**Published:** 2024-02-22

**Authors:** Yanjiao Zuo, Hao Sun, Hongming Pan, Ruihu Zhao, Yingwei Xue, Hongjiang Song

**Affiliations:** Harbin Medical University Cancer Hospital, Harbin Medical University, 150 Haping Road, Nangang District, Harbin 150081, China; 202201502@hrbmu.edu.cn (Y.Z.); haosun@hrbmu.edu.cn (H.S.); 202101465@hrbmu.edu.cn (H.P.); 2020021716@hrbmu.edu.cn (R.Z.); xueyingwei@hrbmu.edu.cn (Y.X.)

**Keywords:** gastric cancer, immunotherapy, inflammatory and nutritional markers, gastric cancer immune prognostic score, prognostic factor

## Abstract

**Simple Summary:**

In the current landscape of gastric cancer treatment, identifying patients who can benefit from immunotherapy and determining high-risk individuals post immunotherapy remain pivotal objectives. Numerous studies have confirmed the value of classical inflammatory and nutritional markers in predicting the prognosis of patients undergoing immunotherapy. However, these markers have been in use for a long time, and the treatment strategies for gastric cancer have undergone significant changes. Therefore, proposing a novel biomarker based on the latest treatment strategies is necessary. This study established the Gastric Cancer Immune Prognostic Score (GCIPS) through comprehensive blood parameter analysis before immunotherapy, utilizing Cox regression analysis. Comprising white blood cells, lymphocytes, and the international normalized ratio (INR), the GCIPS not only demonstrated excellent performance in survival analyses across all subgroups but was also identified as an independent prognostic factor in this study. Furthermore, the GCIPS exhibited the highest prognostic value, surpassing even the TNM stage and radical resection. Analysis in the validation set further confirmed the accuracy and stability of the results. The proposal of the GCIPS provides a new reference for developing immunotherapy strategies.

**Abstract:**

(1) Background: This study aims to explore the predictive capability of the Gastric Cancer Immune Prognostic Score (GCIPS) for an unfavorable prognosis in gastric cancer patients undergoing immune checkpoint inhibitor (ICI) treatment. (2) Methods: This study included 302 gastric cancer patients who underwent treatment with ICIs at our institution from January 2017 to December 2022. The patients were randomly divided into a test set (201 cases) and a validation set (101 cases) using a random number table. Kaplan–Meier survival analysis and the log-rank test were used to investigate survival differences. Cox regression analysis and Lasso regression analysis were employed to establish the GCIPS and identify independent prognostic indicators. ROC curves, time–ROC curves, and nomograms were utilized to further explore the predictive performance of GCIPS. (3) Results: The test set and validation set showed no statistical differences in clinical and pathological features, as well as blood parameters (all *p* > 0.05). Cox regression analysis revealed that white blood cells (WBC), lymphocytes (LYM), and the international normalized ratio (INR) emerged as independent prognostic blood indicators after eliminating collinearity through Lasso analysis. The GCIPS was established using β coefficients with the following formula: GCIPS = WBC (10^9^/L) × 0.071 − LYM (10^9^/L) × 0.375 + INR × 2.986. ROC curves based on death and time–ROC curves demonstrated that the GCIPS had higher AUCs than other classical markers at most time points. Survival analyses of all subgroups also revealed a significant correlation between the GCIPS and patients’ progression-free survival (PFS) and overall survival (OS) (all *p* < 0.05). Furthermore, the GCIPS was identified as an independent prognostic factor for both PFS and OS. Analyses in the validation set further confirmed the reliability and stability of the GCIPS in predicting patient prognosis. Finally, nomograms incorporating the GCIPS exhibited high accuracy in both the test and validation sets. Additionally, the nomograms revealed that the GCIPS had a higher prognostic value than any other factor, including the TNM stage. (4) Conclusions: The GCIPS demonstrated its ability to predict adverse outcomes in gastric cancer patients undergoing ICIs treatment and had a high prognostic value. As a readily accessible and simple novel biomarker, it effectively identified high-risk patients.

## 1. Introduction

As the fifth most common cancer, gastric cancer remains a major global health concern and a longstanding focal point of medical research [1]. While traditional treatment methods have somewhat improved patient survival rates, their effectiveness remains limited [2,3]. In recent years, immunotherapy has emerged as an innovative therapeutic approach, offering new hope for those affected by gastric cancer [4,5,6]. This treatment activates the patient’s immune system, enabling it to actively identify and eliminate tumor cells [7,8]. While immunotherapy holds significant potential, it is not universally effective for all patients. In gastric cancer, only a small subset of patients demonstrates sensitivity to immune checkpoint inhibitors (ICIs). Additionally, currently available biomarkers such as PD-1/PD-L1 expression and Microsatellite Instability (MSI) not only incur high costs but also fail to comprehensively cover patients who may benefit from ICIs. This underscores the urgent need for a reliable biomarker to accurately predict patient responses and provide robust support for treatment strategies [9,10].

Non-invasive biomarkers have gained widespread attention due to their simplicity, accessibility, and relative accuracy in the field of immunotherapy [11,12,13]. Previous research has confirmed the predictive value of traditional inflammatory and nutritional markers in forecasting the prognosis of immunotherapy patients [14,15,16]. Parameters such as the prognostic nutritional index (PNI), neutrophil-to-lymphocyte ratio (NLR), platelet-to-lymphocyte ratio (PLR), monocyte-to-lymphocyte ratio (MLR), systemic immune-inflammation index (SII), and systemic immune response index (SIRI) have been identified to correlate with the effectiveness of immunotherapy [17,18,19,20]. Liu et al. conducted an analysis of multiple classical indicators for their predictive abilities in hepatocellular carcinoma (HCC). After analyzing data from 151 HCC patients, they found that these inflammatory and nutritional indicators possess certain prognostic value, particularly the nutritional biomarkers. In a prospective study on gastric cancer, Ding and his colleagues, through the analysis of data from 30 gastric cancer patients, identified that the combination of SII and PNI effectively predicts the outcome for gastric cancer patients receiving sintilimab [17,21]. However, these markers have been established and applied for an extended period. On the other hand, with the advancement of medical technology, the treatment strategy for gastric cancer has evolved from singular approaches to a comprehensive combination of surgery, chemotherapy, targeted therapy, immunotherapy, and psychological treatment [22]. This integrated treatment strategy has significantly prolonged the survival period for patients. In this context, the overall systemic condition of the patients becomes particularly crucial. Given the continuous evolution of treatment strategies for gastric cancer, traditional markers may exhibit limitations when confronted with new treatment modalities and approaches [23]. Therefore, there is an urgent need to introduce a new biomarker based on the latest treatment methods to assess patient responses to immunotherapy more accurately and predict their prognosis.

In this context, we enrolled 302 gastric cancer patients undergoing immunotherapy and, through comprehensive analysis, developed a novel evaluation indicator called the Gastric Cancer Immune Prognostic Score (GCIPS). This scoring system aims to provide a fresh feasible biomarker for a more accurate and reliable assessment of immunotherapy prognosis for gastric cancer patients.

## 2. Materials and Methods

### 2.1. Patients

This study included 302 patients with gastric cancer, all of whom underwent ICI treatment at Harbin Medical University Cancer Hospital between January 2017 and December 2022. Incomplete clinical records, loss to follow-up, and the presence of multiple cancers and other chronic diseases were exclusion criteria for this study. All experimental designs in this study comply with the Helsinki Declaration and its amendments, and the study has received approval from the Ethics Committee of Harbin Medical University Cancer Hospital (Ethics Number: 2019-57-IIT, approved on 17 April 2019).

### 2.2. Data Collection and Follow-Up

To minimize potential bias, we employed a random number table to divide the patients into a training set (*n* = 201) and a validation set (*n* = 101). We retrospectively collected clinical and medical records for all patients using the medical record system. Additionally, to establish the GCIPS, we gathered the results of all pre-treatment blood tests performed on the patients. The primary endpoints of this study were progression-free survival (PFS) and overall survival (OS), obtained through regular telephone follow-ups over a follow-up period of 73.27 months. Specifically, PFS was defined as the time from the start of treatment to disease progression, death, or the last follow-up. Disease progression was confirmed through comprehensive imaging or pathological examinations. Overall survival was defined as the time from the start of treatment to death or the last follow-up.

### 2.3. Immune Checkpoint Inhibitors

All patients underwent multiple cycles of ICIs treatment, with 203 patients participating in three clinical trials receiving camrelizumab (Clinical Trial Registration Numbers: CTR20200708, approved on 21 April 2020; CTR20200045, approved on 9 January 2020; CTR20190072, approved on 24 January 2019), and 61 patients participating in another clinical trial receiving toripalimab (Clinical Trial Registration Number: CTR20212739, approved on 30 November 2021). The remaining 38 patients, who did not enroll in clinical trials, voluntarily opted for various ICIs, including toripalimab, pembrolizumab, camrelizumab, and sintilimab.

### 2.4. Statistical Analysis

This study defined statistical significance as a bilateral *p*-value of <0.05, and all statistical analyses were conducted using SPSS 25 (Chicago, IL, USA, https://www.ibm.com, accessed on 1 October 2023), R 4.2.3 (Vienna, Austria, https://cran.r-project.org, accessed on 2 October 2023), and GraphPad Prism 8 (San Diego, CA, USA, https://www.graphpad.com, accessed on 1 October 2023). Continuous variables conforming to a Gaussian distribution are presented as mean and standard deviation (SD) and were analyzed for differences using independent samples *t*-tests. Non-normally distributed continuous variables are represented by the median and interquartile range (IQR) and were compared using the Mann–Whitney U test. Categorical variables are expressed as counts and percentages (%), and differences were assessed through chi-square tests or Fisher’s exact tests. We implemented a random number table method to group patients and employed Cox univariate and multivariate regression analysis to establish the GCIPS and identify independent prognostic factors, with the relative risk represented by the hazard ratio (HR) and 95% confidence interval (CI). Additionally, we utilized Least Absolute Shrinkage and Selection Operator (Lasso) regression analysis to alleviate potential multicollinearity. Lasso regression analysis is a statistical method used for feature selection and regression analysis. It achieves sparsity of unimportant variables in the model by penalizing model parameters, effectively addressing multicollinearity. The key feature of Lasso regression is its ability to shrink the coefficients of predictive variables with minimal impact on the target variable to zero, thus facilitating feature selection. By adjusting the regularization parameters (λ value), Lasso regression can identify variables that significantly contribute to the predictive variables, enhancing the model’s generalization ability and interpretability. Then, we evaluated the optimal cutoff value and assessed the prognostic significance of different factors using ROC curves, time–ROC curves, and the area under the curve (AUC). Next, we analyzed survival differences by examining Kaplan–Meier survival curves and conducting log-rank tests. Additionally, we delved into the impact of relevant indicators on survival via proportional risk hypothesis testing, nomograms, and calibration curves.

## 3. Results

### 3.1. Patient Characteristics

In the overall dataset, there were a total of 302 patients, including 200 (66.2%) males and 102 (33.8%) females, with a mean age and body mass index (BMI) of 63.73 (10.56) years and 21.93 (3.28), respectively. This study only included patients in TNM stage III and IV, with 59.9% of them being in stage IV. Due to the rapid progression of the disease, only 177 (58.6%) individuals underwent surgery, among whom only 108 (35.8%) underwent radical resection. There were no differences in clinical and pathological information between the test set and validation set in all patients (all *p* > 0.05, Table 1). To establish the GCIPS, we collected pre-treatment blood test indicators from patients (detailed information is available in Table 2). There were also no differences in blood indicators between the test set and the validation set (*p* > 0.05).

### 3.2. Establishment of the GCIPS in the Test Set

To identify independent prognostic blood parameters influencing OS, we included all blood parameters as continuous variables in a Cox regression analysis. The results showed that γ-glutamyl transferase (γ-GGT), total protein (TP), albumin (ALB), albumin/globulin ratio (A/G), prealbumin (PALB), alkaline phosphatase (ALP), white blood cell count (WBC), neutrophil count (NEU), lymphocyte count (LYM), platelet count (PLT), and the international normalized ratio (INR) were all correlated with OS (all *p* < 0.05). Simultaneously, to prevent the impact of multicollinearity on the results, we conducted a Lasso regression analysis on these indicators before the multivariate analysis. After 303 cycles of validation, the optimal λ was determined to be 0.008, and ALB was excluded due to multicollinearity (Figure 1). After incorporating the remaining indicators associated with OS into the multivariate analysis, it was found that WBC (HR = 1.073, *p* < 0.001), LYM (HR = 0.720, *p* = 0.030), and INR (HR = 1.732, *p* = 0.007) were identified as independent prognostic factors for OS (Table 3).

After incorporating WBC, LYM, and INR into the Cox multivariate model again, the obtained β coefficients were 0.071, −0.375, and 2.986, respectively (Table 4). Therefore, the final formula for calculating the GCIPS was defined as follows: GCIPS = WBC (10^9^/L) × 0.071 − LYM (10^9^/L) × 0.375 + INR × 2.986.

### 3.3. The Prognostic Value of the GCIPS

To explore the prognostic predictive ability of GCIPS, we also calculated several classic inflammatory and nutritional markers in the test set and compared their prognostic value with that of the GCIPS (Table 5). We first calculated the area under the curve (AUC) of all markers by plotting ROC curves based on OS-related deaths (Figure 2). The AUCs of NLR, PLR, MLR, SII, SIRI, PNI, and the GCIPS were 0.581, 0.540, 0.513, 0.544, 0.507, 0.591, and 0.634, respectively, with the GCIPS exhibiting the highest AUC (Table 6).

Additionally, we generated time–ROC curves for the GCIPS and compared their predictive ability at different time points with other markers. The AUCs for the GCIPS in predicting PFS at 3, 4, and 5 years were 0.703, 0.724, and 0.786, respectively, and for OS, they were 0.699, 0.724, and 0.798, demonstrating consistently high levels of prediction (Figure 3A,B). In the comparative analysis of time–ROC curves at different time points, the GCIPS consistently demonstrated a leading position, both in PFS and OS (Figure 3C,D).

### 3.4. Survival Analysis of the GCIPS in the Test Set

#### 3.4.1. Cox Regression Analysis

We conducted a Cox regression analysis for the GCIPS and other pathological factors. The results showed that the GCIPS, subclavian lymph nodes (SLN), surgery, radical resection, tumor size, TNM stage, CEA, CA199, and CA724 were associated with PFS (all *p* < 0.05). After including them in the Cox multivariate analysis, the GCIPS (HR = 2.020, *p* = 0.009), radical resection (HR = 1.922, *p* = 0.010), TNM stage (HR = 1.800, *p* = 0.048), and CA724 (HR = 1.494, *p* = 0.046) were identified as independent prognostic factors for PFS (Table 7). Furthermore, analysis for OS revealed that the GCIPS, SLN, surgery, radical resection, TNM stage, CEA, CA199, and CA724 were associated with patient survival (all *p* < 0.05). The GCIPS (HR = 2.272, *p* < 0.001), radical resection (HR = 1.901, *p* = 0.011), and TNM stage (HR = 1.755, *p* = 0.0404) were also identified as independent prognostic factors influencing survival (Table 8).

#### 3.4.2. Kaplan–Meier Survival Analysis

We plotted the survival curves for the GCIPS in the test set and observed that patients with a higher GCIPS had a poorer PFS (χ^2^ = 7.375, *p* = 0.007) and OS (χ^2^ = 12.277, *p* < 0.001, Figure 4A,B). Additionally, given that radical resection and TNM stage were identified as independent prognostic factors for PFS and OS, we conducted subgroup analyses of patients with different surgery and TNM stages. A total of 74 patients (low GCIPS = 28, high GCIPS = 46) were diagnosed at TNM stage III. Notably, those with higher GCIPS exhibited significantly shorter PFS and OS (χ^2^ = 6.042, *p* = 0.014, and χ^2^ = 8.554, *p* = 0.003, Figure 5A,B). Among the 127 patients (low GCIPS = 36, high GCIPS = 91) in TNM stage IV, individuals with higher GCIPS similarly demonstrated shorter PFS and OS (χ^2^ = 5.421, *p* = 0.047, and χ^2^ = 8.553, *p* = 0.004, Figure 5C,D). Moreover, among the 65 patients (low GCIPS = 23, high GCIPS = 42) who underwent radical resection, a notable correlation was observed between higher GCIPS and shorter PFS and OS (χ^2^ = 5.664, *p* = 0.017, and χ^2^ = 8.709, *p* = 0.003, Figure 6A,B). Similarly, in the case of the 136 patients (low GCIPS = 41, high GCIPS = 95) who did not undergo radical resection, higher GCIPS was associated with shorter PFS and OS (χ^2^ = 5.302, *p* = 0.032, and χ^2^ = 6.770, *p* = 0.011, Figure 6C,D).

### 3.5. Survival Analysis of the GCIPS in the Validation Set

To further validate its reliability, we reanalyzed the GCIPS in the validation set using the same cutoff value. The AUCs of the GCIPS for PFS at 3, 4, and 5 years were 0.663, 0.734, and 0.787, respectively. For OS, the AUCs were 0.699, 0.714, and 0.763, also indicating high levels of predictive ability (Figure 7A,B). Meanwhile, the GCIPS also showed a significant negative correlation with PFS (χ^2^ = 6.013, *p* = 0.014) and OS (χ^2^ = 11.012, *p* < 0.001) in the validation set (Figure 7C,D).

### 3.6. Nomograms of the GCIPS

We conducted a proportional hazards assumption test on the independent prognostic factors in this study and found that they did not violate the proportional hazards assumption (Figure 8A,B). Finally, based on the results of the multivariate analysis, we created nomograms for PFS and OS in the test set, and the C-index of the nomograms was 0.719 and 0.742, respectively (Figure 8C,D). Additionally, we analyzed the predictive performance of the nomograms using the validation set. The C-index of the nomograms in the validation set was 0.699 and 0.713, respectively, indicating a high level of accuracy. The calibration curves drawn in the validation set also demonstrated the good predictive accuracy of the nomograms (Figure 8E,F).

## 4. Discussion

Since the application of ICIs in gastric cancer, there has been significant interest in the value of identifying effective predictive biomarkers for ICIs. In 2022, Nose and colleagues collected blood samples from 29 gastric cancer patients undergoing ICI treatment. They utilized flow cytometry to investigate the relationship between the frequency of CD103 in PD-1-CD8 T cells and patients’ PFS. The results revealed a significant correlation between a higher frequency of CD103 in PD-1-CD8 T cells and longer PFS, indicating the ability to predict the efficacy of anti-ICI treatment [24]. Yang and his colleagues established a predictive model for immunotherapy by downloading data from The Cancer Genome Atlas (TCGA) and Gene Expression Omnibus (GEO). They selected six lactylation-related genes to create a lactylation score, finding that it could predict the immune progression and immune escape in gastric cancer. At the same time, they also found that this model is associated with the response to ICIs, serving as a potential biomarker for the efficacy of ICI treatment [25]. Non-invasive biomarkers based on blood indicators have also garnered attention due to their simplicity and ready availability. In 2022, Sun and colleagues explored the application of the PNI in immunotherapy. They collected data from 146 gastric cancer patients and analyzed the impact of the PNI on PFS and OS. The results demonstrated that the PNI exhibited higher prognostic value in all subgroup analyses [26]. Zhang and his colleagues’ meta-analysis further corroborated these findings, integrating data and results from 17 studies, revealing that the PNI is a reliable predictive factor for gastric cancer patients receiving ICIs [27]. Meanwhile, Wan and colleagues have extensively studied inflammatory markers and found that the NLR was associated with the benefits of immunotherapy [28]. On the other hand, there is a growing awareness of the necessity to establish new biomarkers specifically for immunotherapy. In 2018, Mezquita collected data from 466 advanced non-small cell lung cancer patients across eight centers and analyzed their inflammatory status. Ultimately, they established a new lung immune prognostic index (LIPI) and found its prognostic value to be higher in patients treated with ICIs compared to those undergoing chemotherapy [29]. This clearly illustrated the advantages of establishing new indicators based on patients treated with ICIs.

In this study, we established the independent blood parameters affecting patients in the test set through a Cox regression analysis and a Lasso regression analysis, identifying WBC, LYM, and INR. Based on this, we constructed the GCIPS. After comparing the predictive abilities of the GCIPS with other classical biomarkers, we found that the GCIPS consistently maintained the highest AUC at most time points, indicating a superior prognostic predictive capability. The GCIPS also demonstrated a significant correlation with survival in all patient groups and different subgroups. Additionally, GCIPS, TNM stage, and radical resection were identified as independent prognostic factors for both PFS and OS. The stability of the GCIPS in predicting patient prognosis was confirmed by time–ROC curves and survival curves in the test set. Finally, nomograms incorporating the GCIPS not only exhibited a high accuracy but also revealed that the GCIPS’s prognostic value surpassed even the TNM stage and radical resection in the prediction model. This further confirms the significant advantages of GCIPS in patients who received ICIs.

The GCIPS, composed of WBC, LYM, and INR, was found to predict the survival of cancer patients [30,31,32]. The WBC count encompasses various cell subtypes, including NEU, LYM, monocytes (MON), eosinophils (EOS), and basophils (BAS) [33]. Except for EOS and BAS, which constituted a smaller proportion, the remaining cell subtypes played crucial roles in the initiation and development of tumors [34,35]. NEU, the most common type of WBC, constituting 60–70%, played a vital role in infections and inflammation [36]. However, in the tumor microenvironment, NEU exhibited a dual effect [37]. On the one hand, they participated in the engulfment and destruction of tumors through the release of inflammatory factors and the generation of reactive oxygen species [38]. On the other hand, various cell factors produced by NEU, such as epidermal growth factor, vascular endothelial growth factor, and transforming growth factor-β (TGF-β), promoted angiogenesis and tumor cell growth [39,40]. Furthermore, certain inhibitory factors secreted by NEU could suppress the activity of lymphocytes, thereby weakening the anti-tumor immune response [41]. MON constituted 2–8% of WBC and could differentiate into macrophages, participating in the engulfment and clearance of cellular debris and pathogens [42]. Like NEU, MON also exhibited a dual effect in tumor progression [43]. Some cell factors secreted by MON, such as interleukin-6, interleukin-10, and TGF-β, could inhibit lymphocyte activity and promote the growth of tumor cells [44,45]. LYM constituted 20–30% of WBC and were divided into subgroups such as T lymphocytes, B lymphocytes, and natural killer cells [46]. Unlike other subgroups, LYM played a crucial role in anti-tumor immunity, especially in immunotherapy [47,48]. ICIs enhance the immune system’s recognition of tumor cells, thereby boosting the anti-tumor immune response [49,50]. Therefore, the efficacy of ICIs depends on the patient’s immune function. A decrease in LYM might indicate a poorer immune status in patients, potentially reducing their response to ICIs [51,52]. Additionally, WBCs, to some extent, reflect the body’s tumor burden, with an overall higher WBC potentially indicating a more active tumor [53]. INR reflects the body’s inflammatory status and coagulation function [54]. An elevated INR indicates that the body is in a state of chronic inflammation and coagulation dysfunction, reflecting not only a high tumor burden but also directly influencing patient survival [55,56]. In addition, INR levels were influenced by nutritional status and liver function, factors closely related to the patient’s overall health and response to treatment [57]. Therefore, INR was defined as an important component of the GCIPS.

In this study, GCIPS created based on pre-treatment blood indicators demonstrated significant prognostic value in patients receiving ICIs. This provides clinicians with an easily accessible biomarker to assess patients who may benefit from ICIs. Additionally, GCIPS may assist clinicians in risk stratification, enabling the identification of high-risk patients for timely intervention. The potential value of GCIPS in clinical practice deserves further exploration.

While this study yielded meaningful findings, several limitations should be noted. Firstly, due to the retrospective design employed in this research, it was challenging to eliminate potential information bias. Secondly, the relatively small sample size of patients in this study might have impacted the generalizability of the results. Further large-scale studies would validate and solidify our research conclusions. Thirdly, the cutoff value for the GCIPS in this study was determined through ROC curve analysis, which might have been influenced to some extent by the sample size. Therefore, the stability of the cutoff value should be further confirmed in studies with a larger scope. Lastly, despite the significant potential demonstrated by the GCIPS in predicting the efficacy of ICIs, this study did not delve into its molecular mechanisms, providing valuable direction for future research in this field.

## 5. Conclusions

The proposed GCIPS demonstrated its ability to predict adverse outcomes in gastric cancer patients undergoing immunotherapy and had a high prognostic value. As a readily accessible and simple novel biomarker, it effectively identified high-risk patients.

## Figures and Tables

**Figure 1 biomedicines-12-00491-f001:**
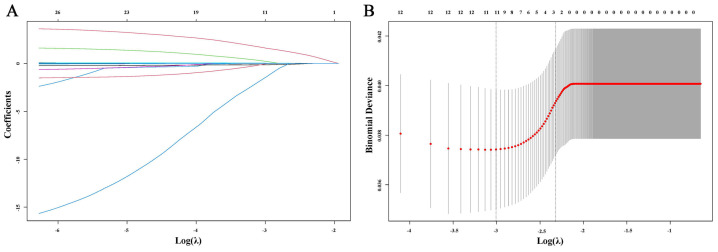
Lasso regression analysis. (**A**) The variation characteristics of the coefficient of variables. (**B**) The selection process of the optimum value of λ.

**Figure 2 biomedicines-12-00491-f002:**
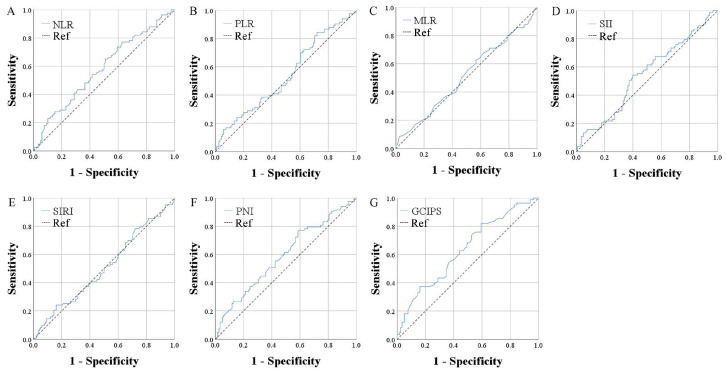
ROC curves of all markers. (**A**) ROC curve of NLR; (**B**) ROC curve of PLR; (**C**) ROC curve of MLR; (**D**) ROC curve of SII; (**E**) ROC curve of SIRI; (**F**) ROC curve of PNI; (**G**) ROC curve of GCIPS. NLR: neutrophil-to-lymphocyte ratio; PLR: platelet-to-lymphocyte ratio; MLR: monocyte-to-lymphocyte ratio; SII: systemic immune-inflammation index; SIRI: systemic inflammation response index; PNI: prognostic nutritional index; GCIPS: Gastric Cancer Immune Prognostic Score.

**Figure 3 biomedicines-12-00491-f003:**
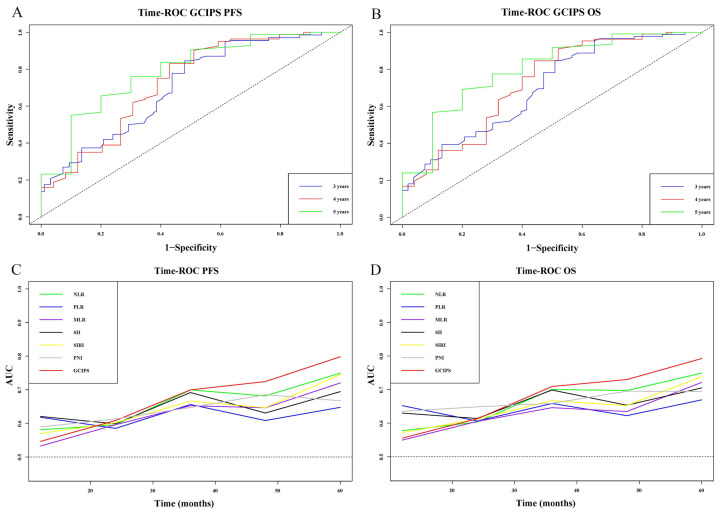
Time–ROC curves for PFS and OS. (**A**) The 3-, 4-, and 5-year time–ROC curves of the GCIPS for PFS. (**B**) The 3-, 4-, and 5-year time–ROC curves of the GCIPS for OS. (**C**) The comparison of time–ROC curves with different markers for PFS. (**D**) The comparison of time–ROC curves with different markers for OS.

**Figure 4 biomedicines-12-00491-f004:**
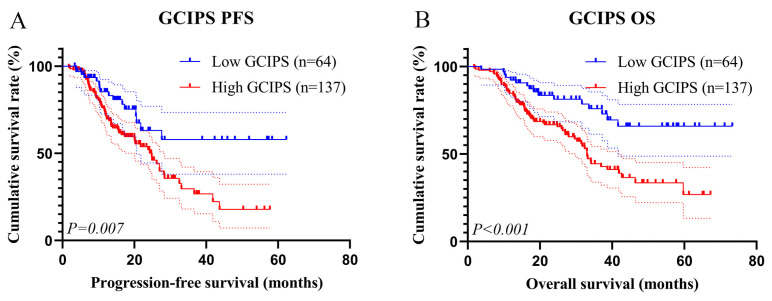
K–M survival curves for GCIPS. (**A**) K–M survival curves of GCIPS for PFS; (**B**) K–M survival curves of GCIPS for OS.

**Figure 5 biomedicines-12-00491-f005:**
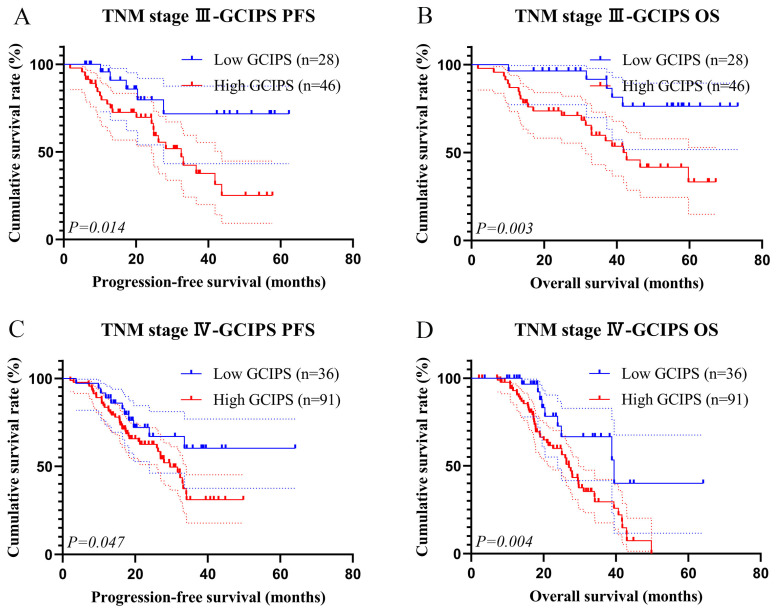
K–M survival curves for GCIPS in different TNM stages. (**A**) K–M survival curves of GCIPS for PFS in TNM stage III; (**B**) K–M survival curves of GCIPS for OS in TNM stage III; (**C**) K–M survival curves of GCIPS for PFS in TNM stage IV; (**D**) K–M survival curves of GCIPS for OS in TNM stage IV.

**Figure 6 biomedicines-12-00491-f006:**
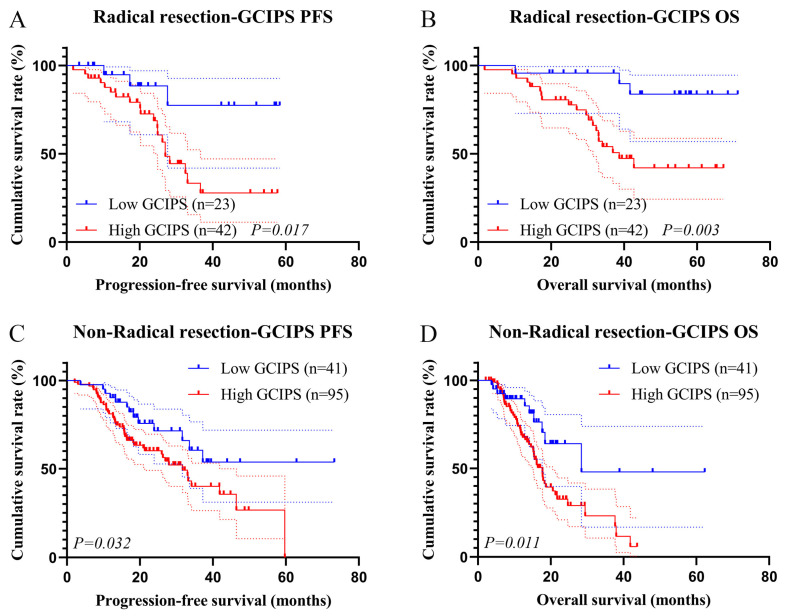
K–M survival curves for GCIPS in radical resection. (**A**) K–M survival curves of GCIPS for PFS in patients with radical resection; (**B**) K–M survival curves of GCIPS for OS in patients with radical resection; (**C**) K–M survival curves of GCIPS for PFS in patients without radical resection; (**D**) K–M survival curves of GCIPS for OS in patients without radical resection.

**Figure 7 biomedicines-12-00491-f007:**
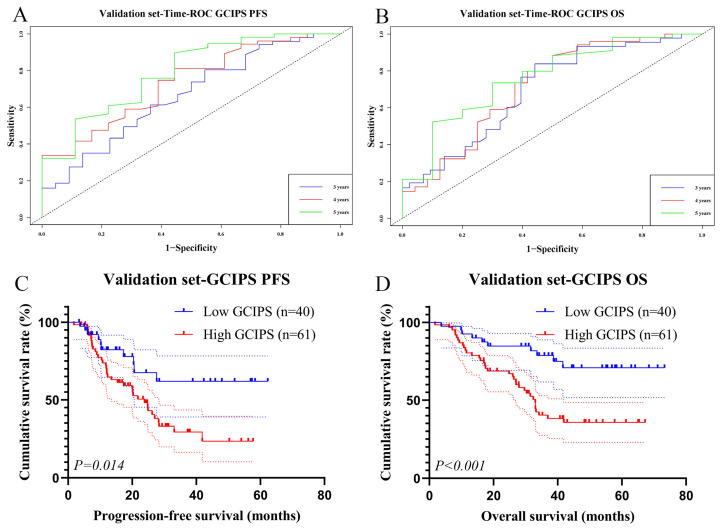
Survival analysis of GCIPS in the validation set. (**A**) Time–ROC of GCIPS for PFS. (**B**) Time–ROC of GCIPS for OS. (**C**) K–M survival curve of GCIPS for PFS. (**D**) K–M survival curve of GCIPS for OS.

**Figure 8 biomedicines-12-00491-f008:**
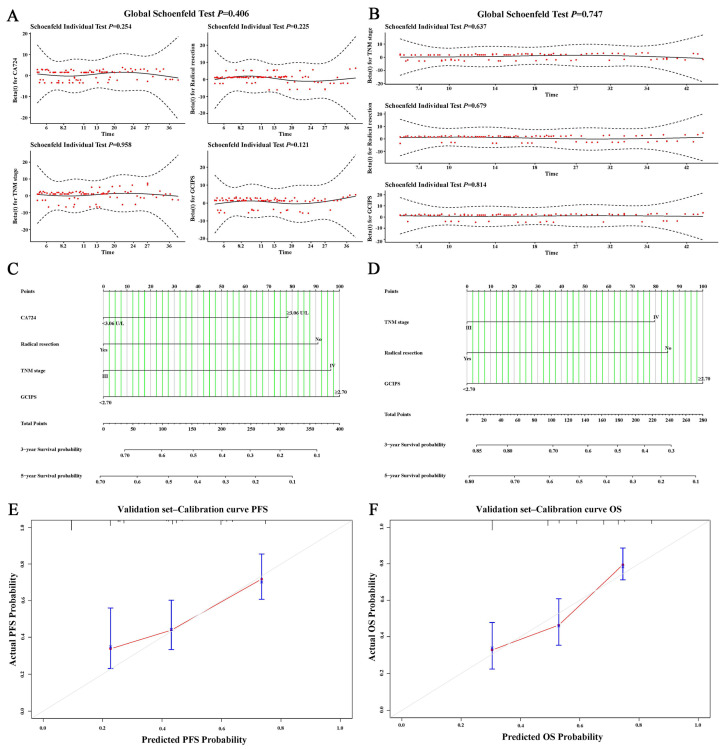
Nomograms for PFS and OS. (**A**) Schoenfeld residual plot for PFS. (**B**) Schoenfeld residual plot for OS. (**C**) Nomogram for PFS. (**D**) Nomogram for OS. (**E**) Calibration curve for PFS. (**F**) Calibration curve for OS.

**Table 1 biomedicines-12-00491-t001:** Patient characteristics.

	Total Set	Test Set	Validation Set	*p*
Items	*n* = 302	*n* = 201	*n* = 101	
Age (years), mean (SD)	63.73 (10.56)	57.94 (9.97)	57.72 (9.94)	0.861
Sex, n (%)				0.316
Male	200 (66.2)	137 (68.2)	63 (62.4)	
Female	102 (33.8)	64 (31.8)	38 (37.6)	
BMI (Kg/m^2^), mean (SD)	21.93 (3.28)	21.95 (3.26)	21.90 (3.34)	0.906
SLN, n (%)				0.443
Positive	36 (11.9)	26 (12.9)	10 (9.9)	
Negative	266 (88.1)	175 (87.1)	91 (90.1)	
Surgery, n (%)				0.092
Yes	177 (58.6)	111 (55.2)	66 (65.3)	
No	125 (41.4)	90 (44.8)	35 (34.7)	
Radical resection, n (%)				0.080
Yes	108 (35.8)	65 (32.3)	43 (42.6)	
No	194 (64.2)	136 (67.7)	58 (57.4)	
Primary tumor site, n (%)				0.912
Upper 1/3	44 (14.6)	30 (14.9)	14 (13.9)	
Middle 1/3	66 (21.9)	45 (22.4)	21 (20.8)	
Low 1/3	184 (60.9)	120 (59.7)	64 (63.4)	
Whole	8 (2.6)	6 (3.0)	2 (2.0)	
Borrmann type, n (%)				0.173
I	38 (12.6)	25 (12.4)	13 (12.9)	
II	6 (2.0)	4 (2.0)	2 (2.0)	
III	193 (63.9)	136 (67.7)	57 (56.4)	
IV	65 (21.5)	36 (17.9)	29 (28.7)	
Tumor size, n (%)				0.952
<20 mm	55 (18.2)	36 (17.9)	19 (18.8)	
20–50 mm	32 (10.6)	22 (10.9)	10 (9.9)	
>50 mm	215 (71.2)	143 (71.1)	72 (71.3)	
Differentiation, n (%)				0.157
Poor	121 (40.1)	74 (36.8)	47 (46.5)	
Moderate	26 (8.6)	15 (7.5)	11 (10.9)	
Good	2 (0.7)	1 (0.5)	1 (1.0)	
Unknown	153 (50.7)	111 (55.2)	42 (41.6)	
Lauren type, n (%)				0.237
Intestinal	27 (8.9)	16 (8.0)	11 (10.9)	
Diffuse	27 (8.9)	16 (8.0)	11 (10.9)	
Mixed	34 (11.3)	19 (9.5)	15 (14.9)	
Unknown	214 (70.9)	150 (74.6)	64 (63.4)	
TNM stage, n (%)				0.104
III	121 (40.1)	74 (36.8)	47 (46.5)	
IV	181 (59.9)	127 (63.2)	54 (53.5)	
AFP, n (%)				0.272
<2.96 ng/mL	151 (50.0)	105 (52.2)	46 (45.5)	
≥2.96 ng/mL	151 (50.0)	96 (47.8)	55 (54.5)	
CEA, n (%)				
<2.43 ng/mL	151 (50.0)	91 (45.3)	60 (59.4)	0.051
≥2.43 ng/mL	151 (50.0)	110 (54.7)	41 (40.6)	
CA199, n (%)				0.080
<14.40 U/L	149 (49.3)	92 (45.8)	57 (56.4)	
≥14.40 U/L	153 (50.7)	109 (54.2)	44 (43.6)	
CA724, n (%)				0.155
<3.06 U/L	150 (49.7)	94 (46.8)	56 (55.4)	
≥3.06 U/L	152 (50.3)	107 (53.2)	45 (44.6)	
CA125Ⅱ, n (%)				0.653
<25.19 U/L	144 (47.7)	94 (46.8)	50 (49.5)	
≥25.19 U/L	158 (52.3)	107 (53.2)	51 (50.5)	

SD: standard deviation; BMI: body mass index; SLN: subclavian lymph nodes; AFP: alpha-fetoprotein; CEA: carcinoembryonic antigen; CA199: carbohydrate antigen 199; CA724: carbohydrate antigen 724; CA125II: carbohydrate antigen 125II.

**Table 2 biomedicines-12-00491-t002:** Blood parameters.

	Total Set	Test Set	Validation Set	*p*
Items	*n* = 302	*n* = 201	*n* = 101	
ALT (U/L, median (IQR))	14.00 (10.00, 26.00)	14.00 (10.00, 25.95)	14.00 (10.00, 26.00)	0.910
AST (U/L, median (IQR))	20.00 (16.00, 27.00)	20.00 (16.00, 28.00)	20.00 (16.00, 25.00)	0.559
γ-GGT (U/L, median (IQR))	23.00 (16.00, 45.50)	24.00 (16.00, 46.00)	21.00 (16.00, 45.50)	0.661
LDH (U/L, median (IQR))	171.00 (147.00, 215.00)	174.00 (148.00, 216.50)	167.00 (144.50, 208.50)	0.164
TBIL (μmol/L, median (IQR))	12.00 (9.20, 15.60)	12.00 (9.20, 15.70)	12.00 (9.15, 15.31)	0.653
DBIL (μmol/L, median (IQR))	2.70 (1.93, 3.66)	2.60 (1.85, 3.60)	2.80 (2.03, 3.86)	0.218
IDBIL (μmol/L, median (IQR))	9.00 (7.00, 12.10)	9.30 (7.10, 12.17)	8.79 (6.63, 11.85)	0.235
TP (g/L, mean (SD))	69.13 (7.10)	69.32 (7.11)	68.73 (7.08)	0.494
ALB (g/L, mean (SD))	38.98 (4.43)	38.85 (4.46)	39.26 (4.40)	0.447
GLOB (g/L, mean (SD))	30.06 (5.11)	30.36 (5.22)	29.46 (4.85)	0.153
A/G, mean (SD)	1.33 (0.25)	1.31 (0.25)	1.35 (0.26)	0.129
PALB (mg/L, mean (SD))	199.60 (65.32)	202.30 (65.66)	194.24 (64.63)	0.313
BUN (mmol/L, mean (SD))	5.52 (1.60)	5.61 (1.62)	5.33 (1.56)	0.158
CREA (μmol/L, mean (SD))	76.14 (16.56)	76.08 (16.15)	76.27 (17.44)	0.926
UA (μmol/L, mean (SD))	290.70 (83.67)	296.76 (87.07)	278.64 (75.44)	0.076
ALP (U/L, median (IQR))	90.00 (73.00, 121.00)	89.00 (71.50, 121.00)	91.00 (73.50, 121.00)	0.631
Glu (mmol/L, median (IQR))	5.10 (4.57, 5.80)	5.10 (4.60, 5.80)	5.10 (4.50, 5.80)	0.513
WBC (10^9^/L, median (IQR))	6.40 (5.07, 8.09)	6.46 (5.13, 8.33)	6.31 (4.90, 7.93)	0.232
NEU (10^9^/L, median (IQR))	3.90 (2.84, 5.38)	3.97 (2.92, 5.44)	3.76 (2.72, 5.03)	0.247
LYM (10^9^/L, median (IQR))	1.65 (1.24, 1.99)	1.65 (1.24, 2.03)	1.63 (1.22, 1.93)	0.601
MON (10^9^/L, median (IQR))	0.48 (0.32, 0.62)	0.49 (0.33, 0.64)	0.44 (0.29, 0.60)	0.085
RBC (10^9^/L, mean (SD))	4.27 (0.62)	4.29 (0.62)	4.23 (0.62)	0.492
HGB (10^9^/L, mean (SD))	122.15 (21.75)	122.70 (22.44)	121.07 (20.39)	0.539
HCT (10^9^/L, mean (SD))	37.86 (6.38)	38.10 (6.81)	37.40 (5.41)	0.337
PLT (10^9^/L, median (IQR))	236.50 (184.00, 312.00)	241.00 (186.50, 314.00)	232.00 (180.50, 308.50)	0.546
INR, mean (SD)	1.00 (0.13)	1.01 (0.12)	0.98 (0.15)	0.135
Fbg (g/L, median (IQR))	3.47 (2.80, 4.36)	3.47 (2.80, 4.37)	3.47 (2.79, 4.34)	0.612
Ddi (ng/L, median (IQR))	0.75 (0.41, 1.58)	0.67 (0.41, 1.19)	0.78 (0.46, 1.80)	0.211

IQR: interquartile range; SD: standard deviation; ALT: alanine transaminase; AST: aspartate aminotransferase; γ-GGT: γ-glutamyl transferase; LDH: lactate dehydrogenase; TBIL: total bilirubin; DBIL: direct bilirubin; IDBIL: indirect bilirubin; TP: total protein; ALB: albumin; GLOB: globulin; PALB: prealbumin; BUN: blood urea nitrogen; CREA: creatinine; UA: uric acid; ALP: alkaline phosphatase; Glu: glucose; WBC: white blood cells; NEU: neutrophils; LYM: lymphocytes; MON: monocytes; RBC: red blood cells; HGB: hemoglobin; HCT: hematocrit; PLT: platelets; INR: international normalized ratio; Fbg: fibrinogen; Ddi: D-dimer.

**Table 3 biomedicines-12-00491-t003:** Cox regression analysis.

	Univariate Analysis			Multivariate Analysis		
Items	HR	95% CI	*p*	HR	95% CI	*p*
ALT (U/L)	0.996	0.985–1.008	0.509			
AST (U/L)	1.008	0.999–1.017	0.070			
γ-GGT (U/L)	1.002	1.000–1.004	0.027	1.001	0.998–1.003	0.543
LDH (U/L)	1.001	1.000–1.002	0.068			
TBIL (μmol/L)	1.004	0.999–1.008	0.106			
DBIL (μmol/L)	1.007	0.999–1.016	0.089			
IDBIL (μmol/L)	1.007	0.998–1.016	0.141			
TP (g/L)	0.977	0.955–0.999	0.044	0.978	0.954–1.002	0.077
ALB (g/L)	0.942	0.908–0.978	0.002			
GLOB (g/L)	0.997	0.964–1.031	0.862			
A/G	0.452	0.219–0.932	0.032	0.553	0.243–1.260	0.159
PALB (mg/L)	0.995	0.992–0.998	0.001	0.998	0.994–1.001	0.174
BUN (mmol/L)	0.973	0.862–1.097	0.651			
CREA (μmol/L)	0.993	0.981–1.004	0.216			
UA (μmol/L)	0.999	0.996–1.001	0.172			
ALP (U/L)	1.002	1.000–1.004	0.022	1.001	0.999–1.003	0.399
Glu (mmol/L)	0.943	0.813–1.094	0.439			
WBC (10^9^/L)	1.084	1.046–1.123	<0.001	1.073	1.035–1.113	<0.001
NEU (10^9^/L)	1.083	1.006–1.166	0.034	0.951	0.871–1.037	0.256
LYM (10^9^/L)	0.637	0.475–0.854	0.003	0.720	0.535–0.969	0.030
MON (10^9^/L)	0.830	0.355–1.940	0.667			
RBC (10^9^/L)	0.798	0.600–1.061	0.120			
HGB (10^9^/L)	0.997	0.989–1.005	0.425			
HCT (10^9^/L)	0.992	0.965–1.020	0.567			
PLT (10^9^/L)	1.002	1.000–1.004	0.026	1.002	1.000–1.004	0.127
INR	2.788	1.748–3.335	<0.001	1.732	1.034–3.812	0.007
Fbg (g/L)	1.017	0.921–1.124	0.739			
Ddi (ng/L)	1.066	0.967–1.175	0.198			

HR: hazard ratio; CI: confidence interval; ALT: alanine transaminase; AST: aspartate aminotransferase; γ-GGT: γ-glutamyl transferase; LDH: lactate dehydrogenase; TBIL: total bilirubin; DBIL: direct bilirubin; IDBIL: indirect bilirubin; TP: total protein; ALB: albumin; GLOB: globulin; PALB: prealbumin; BUN: blood urea nitrogen; CREA: creatinine; UA: uric acid; ALP: alkaline phosphatase; Glu: glucose; WBC: white blood cells; NEU: neutrophils; LYM: lymphocytes; MON: monocytes; RBC: red blood cells; HGB: hemoglobin; HCT: hematocrit; PLT: platelets; INR: international normalized ratio; Fbg: fibrinogen; Ddi: D-dimer.

**Table 4 biomedicines-12-00491-t004:** Multivariate analysis for WBC, LYM, and INR.

Items	β Value	HR	95% CI	*p*
WBC (10^9^/L)	0.071	1.074	1.038–1.111	<0.001
LYM (10^9^/L)	−0.375	0.687	0.512–0.922	0.012
INR	2.986	2.809	1.882–3.132	0.002

HR: hazard ratio; CI: confidence interval; WBC: white blood cells; LYM: lymphocytes; INR: international normalized ratio.

**Table 5 biomedicines-12-00491-t005:** The calculation formulas.

Items	Calculation Formulas
NLR	Neutrophils (10^9^/L)/lymphocytes (10^9^/L)
PLR	Platelets (10^9^/L) /lymphocytes (10^9^/L)
MLR	Monocytes (10^9^/L)/lymphocytes (10^9^/L)
SII	Platelets (10^9^/L) × neutrophils (10^9^/L)/lymphocytes (10^9^/L)
SIRI	Monocytes (10^9^/L) × neutrophils (10^9^/L)/lymphocytes (10^9^/L)
PNI	Albumin (g/dL) + 5 × lymphocytes (10^9^/L)

NLR: Neutrophil-to-lymphocyte ratio; PLR: platelet-to-lymphocyte ratio; MLR: monocyte-to-lymphocyte ratio; SII: systemic immune-inflammation index; SIRI: systemic inflammation response index; PNI: prognostic nutritional index.

**Table 6 biomedicines-12-00491-t006:** AUC values of all markers.

Items	AUC	95% CI
NLR	0.581	0.501–0.661
PLR	0.540	0.459–0.621
MLR	0.513	0.431–0.595
SII	0.544	0.463–0.625
SIRI	0.507	0.425–0.588
PNI	0.591	0.511–0.671
GCIPS	0.634	0.557–0.712

AUC: area under the curve; CI: confidence interval; NLR: neutrophil-to-lymphocyte ratio; PLR: Platelet-to-Lymphocyte Ratio; MLR: Monocyte-to-Lymphocyte Ratio; SII: systemic immune-inflammation index; SIRI: systemic inflammation response index; PNI: prognostic nutritional index; GCIPS: Gastric Cancer Immune Prognostic Score.

**Table 7 biomedicines-12-00491-t007:** Cox regression analysis for PFS.

		PFS		
	Univariate Analysis		Multivariate Analysis	
Items	HR (95% CI)	*p*	HR (95% CI)	*p*
Age (years)	1.001 (0.984–1.019)	0.879		
Sex				
Male	Ref			
Female	1.017 (0.708–1.461)	0.927		
BMI (Kg/m^2^)	0.977 (0.927–1.029)	0.376		
GCIPS				
<2.70	Ref		Ref	
≥2.70	2.197 (1.425–3.387)	<0.001	2.020 (1.092–2.708)	0.009
SLN				
Negative	Ref		Ref	
Positive	1.661 (1.037–2.662)	0.035	1.372 (0.842–2.235)	0.204
Surgery				
Yes	Ref		Ref	
No	1.836 (1.275–2.643)	0.001	1.321 (0.811–2.152)	0.264
Radical resection				
Yes	Ref		Ref	
No	2.504 (1.692–3.705)	<0.001	1.922 (1.171–3.156)	0.010
Primary tumor site				
Low 1/3	Ref			
^#^Other	1.004 (0.698–1.444)	0.984		
Borrmann type				
Ⅰ + Ⅱ	Ref			
III + IV	1.062 (0.675–1.671)	0.795		
Tumor size				
<50 mm	Ref		Ref	
≥50 mm	1.513 (1.015–2.254)	0.042	1.087 (0.642–1.842)	0.756
TNM stage				
III	Ref		Ref	
IV	2.810 (1.891–4.175)	<0.001	1.800 (1.005–3.224)	0.048
AFP				
<2.96 ng/mL	Ref			
≥2.96 ng/mL	1.230 (0.869–1.740)	0.242		
CEA				
<2.43 ng/mL	Ref		Ref	
≥2.43 ng/mL	1.801 (1.256–2.582)	0.001	1.322 (0.892–1.960)	0.164
CA199				
<14.40 U/L	Ref		Ref	
≥14.40 U/L	1.326 (0.936–1.879)	0.013	1.079 (0.754–1.546)	0.676
CA724				
<3.06 U/L	Ref		Ref	
≥3.06 U/L	2.210 (1.535–3.180)	<0.001	1.494 (1.007–2.218)	0.046
CA125Ⅱ				
<25.19 U/L	Ref			
≥25.19 U/L	1.229 (0.868–1.739)	0.245		

^#^Others: upper 1/3 + middle 1/3 + whole; PFS: progression-free survival; HR: hazard ratio; CI: confidence interval; BMI: body mass index; SLN: subclavian lymph nodes; AFP: alpha-fetoprotein; CEA: carcinoembryonic antigen; CA199: carbohydrate antigen 199; CA724: carbohydrate antigen 724; CA125II: carbohydrate antigen 125II; GCIPS: gastric cancer immune prognostic score.

**Table 8 biomedicines-12-00491-t008:** Cox regression analysis for OS.

		OS		
	Univariate Analysis		Multivariate Analysis	
Items	HR (95% CI)	*p*	HR (95% CI)	*p*
Age (years)	0.999 (0.982–1.016)	0.880		
Sex				
Male	Ref			
Female	1.078 (0.750–1.549)	0.684		
BMI (Kg/m^2^)	0.973 (0.922–1.026)	0.312		
GCIPS				
<2.70	Ref		Ref	
≥2.70	2.776 (1.801–4.278)	<0.001	2.272 (1.464–3.524)	<0.001
SLN				
Negative	Ref		Ref	
Positive	2.011 (1.254–3.223)	0.004	1.612 (0.995–2.612)	0.053
Surgery				
Yes	Ref		Ref	
No	1.637 (1.144–2.342)	0.007	1.469 (0.925–2.418)	0.101
Radical resection				
Yes	Ref		Ref	
No	2.384 (1.614–3.521)	<0.001	1.901 (1.161–3.113)	0.011
Primary tumor site				
Low 1/3	Ref			
^#^Other	1.001 (0.696–1.439)	0.997		
Borrmann type				
Ⅰ + Ⅱ	Ref			
III + IV	1.229 (0.783–1.930)	0.370		
Tumor size				
<50 mm	Ref			
≥50 mm	1.428 (0.960–2.125)	0.078		
TNM stage				
III	Ref		Ref	
IV	2.279 (1.560–3.330)	<0.001	1.755 (1.201–2.565)	0.004
AFP				
<2.96 ng/mL	Ref			
≥2.96 ng/mL	1.148 (0.812–1.623)	0.435		
CEA				
<2.43 ng/mL	Ref		Ref	
≥2.43 ng/mL	2.081 (1.454–2.979)	<0.001	1.481 (1.000–2.194)	0.050
CA199				
<14.40 U/L	Ref		Ref	
≥14.40 U/L	1.550 (1.092–2.201)	0.014	1.080 (0.751–1.553)	0.678
CA724				
<3.06 U/L	Ref		Ref	
≥3.06 U/L	2.464 (1.713–3.546)	<0.001	1.475 (0.884–2.461)	0.137
CA125Ⅱ				
<25.19 U/L	Ref			
≥25.19 U/L	1.109 (0.784–1.567)	0.560		

^#^Others: upper 1/3 + middle 1/3 + whole; OS: overall survival; HR: hazard ratio; CI: confidence interval; BMI: body mass index; SLN: subclavian lymph nodes; AFP: alpha-fetoprotein; CEA: carcinoembryonic antigen; CA199: carbohydrate antigen 199; CA724: carbohydrate antigen 724; CA125II: carbohydrate antigen 125II; GCIPS: gastric cancer immune prognostic score.

## Data Availability

The authors promise to provide the original data supporting this study without reservation.

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
