# Peer review of "The Gastric Cancer Immune Prognostic Score (GCIPS) Shows Potential in Predicting an Unfavorable Prognosis for Gastric Cancer Patients Undergoing Immune Checkpoint Inhibitor Treatment"

_biomedicines, 2024, doi:10.3390/biomedicines12030491_

Round 1

Reviewer 1 Report

Comments and Suggestions for Authors

This manuscript retrospectively studied gastric cancer patients and aims to use gastric cancer immune prognostic score (GCIPS) as a reference to predict prognosis of the patients with treatment of immune checkpoint inhibitors. To reach such aim, authors collected data from gastric cancer patients between 2017 and 2022. Although this paper is clearly written for our readers, there are two major concerns:

1.     Immunotherapy methods for cancer patients can be greatly various. For example, the immune response can be induced by various immunomodulators, each of which shows effects in a different way. So your GCIPs might not be applicable to predict some immunomodulator-induced immunotherapy.

2.     The patients with cancer are studied between 2017 and 2022. Since covid 19's influence at that time period, the patients were not treated ICIs in a regulator way. In this manuscript, authors did not consider your GCIPs might not be accurate. Conclusions obtained from covid 19 pandemic situation might not equal to the ones obtained under normal situation.  

Other than the scientific concerns, this manuscript should indicate the abbreviation when they first appear, such as COX (line 17), INR (line 18), TNM (line 21), and PFS (line 41). Overall, this manuscript is clear and authors should address the limitations to make GCIPs prediction robust scientifically.  

Comments on the Quality of English Language

English is fine. 

Author Response

Dear reviewer,

Thank you very much for taking the time to review my manuscript and raise such crucial questions. Similarly, I greatly appreciate your recognition of our work. Your feedback serves as a significant motivation for our efforts.

  1. Immunotherapy methods for cancer patients can be greatly various. For example, the immune response can be induced by various immunomodulators, each of which shows effects in a different way. So your GCIPs might not be applicable to predict some immunomodulator-induced immunotherapy.

Response: Thank you very much for your questions. The development of immune checkpoint inhibitors has been rapid, and currently, there are several drugs available. Our study indeed includes patients who have been treated with different immune checkpoint inhibitors. However, despite the diversity of immune checkpoint inhibitors, their overall mechanisms of action still revolve around enhancing the immune system's recognition and elimination of tumor cells by targeting PD-1, PD-L1, or CTLA-4 receptors. This immune system-dependent anti-tumor effect is closely linked to the overall condition of the patients. Therefore, as a biomarker reflecting the overall immune and inflammatory status of the patients, GCIPS is not greatly influenced by the specific differences in immune checkpoint inhibitors. A substantial body of previous research has also been conducted under these circumstances. Mezquita et al. established a novel Lung Immune Prognostic Index (LIPI) by collecting data from 446 non-small cell lung cancer patients undergoing Immune Checkpoint Inhibitor (ICI) treatment. Their patients received various ICIs, including nivolumab, pembrolizumab, atezolizumab, durvalumab, and durvalumab-ipilimumab. Ultimately, they revealed the significant prognostic value of LIPI, which has garnered widespread recognition [1]. Zheng et al. and Sun et al. conducted research on classical inflammatory and nutritional markers without specifically distinguishing the use of immune checkpoint inhibitors, and also meaningful results were obtained [2, 3].

  1. The patients with cancer are studied between 2017 and 2022. Since covid 19's influence at that time period, the patients were not treated ICIs in a regulator way. In this manuscript, authors did not consider your GCIPs might not be accurate. Conclusions obtained from covid 19 pandemic situation might not equal to the ones obtained under normal situation.

Response: Thank you for your thoughtful inquiry. The reported time frame in our study is calculated based on the actual treatment period for patients. However, it's important to note that our study is not a continuous one. During the COVID-19 period, as a cancer hospital, we significantly reduced treatment activities, which did not substantially impact our research. Furthermore, our patient cohort primarily comes from various clinical trials, with the majority undergoing treatment after the onset of COVID-19. In contrast, patients treated before the COVID-19 outbreak were predominantly those who self-administered immune checkpoint inhibitors (ICIs). Therefore, while our study has a window during the COVID-19 period, it does not significantly affect the validity of our research findings.

Allow me to express my gratitude once again for the work you have done on our manuscript. We are aware of our language issues and have opted for the professional language editing service offered by your journal. If there are any further questions, please do not hesitate to inform us, and we will make every effort to resolve them promptly.

Best wish to you!

Sincerely

Hongjiang Song

Reference

  1. Mezquita, L, Auclin, E, Ferrara, R, et al. Association of the Lung Immune Prognostic Index With Immune Checkpoint Inhibitor Outcomes in Patients With Advanced Non-Small Cell Lung Cancer. JAMA ONCOL. 2018; 4 (3): 351-357. doi: 10.1001/jamaoncol.2017.4771
  2. Zheng, F, Meng, Q, Zhang, L, et al. Prognostic roles of hematological indicators for the efficacy and prognosis of immune checkpoint inhibitors in patients with advanced tumors: a retrospective cohort study. World J Surg Oncol. 2023; 21 (1): 198. doi: 10.1186/s12957-023-03077-8
  3. Sun, H, Chen, L, Huang, R, et al. Prognostic nutritional index for predicting the clinical outcomes of patients with gastric cancer who received immune checkpoint inhibitors. Front Nutr. 2022; 9 1038118. doi: 10.3389/fnut.2022.1038118

Reviewer 2 Report

Comments and Suggestions for Authors

1.This research focused on The Gastric Cancer Immune Prognostic Score (GCIPS) Showed Potential in Predicting an Unfavorable Prognosis for Gastric Cancer Patients who Undergoing Immune Checkpoint Inhibitors, after check the pubmed, there were  not so many articles aboult this topic, so this manuscript was very prospective and significant.

2.This manuscript foucus on clinical problems of gastric cancer, with strong clinical value and importantce,very interesting research, and also met the submission topic of this journal,the results was real and the conclusion was convincing, but some places can be more perfect.

3.  In MATERIALS AND METHODS part, GCIPS not carefully introduced should be in detail.To be fact, I donot understand what was GCIPS. “employed a random number table to divide the patients into a training set (n=201) and a validation set (n=101).” have some references?

4. Can this system GCIPS used for lung cancer or liver cancer?

5. GCIPS was whole score, not only one index score, maybe better for Prognostic tracking, but how to Guiding treatment?  Because during treatment, We may foucus on the main problem.

6.I know now training and validation now was hot but all was  retrospective study,  can you use this system to apply in the clinical medicine really.

7.Some figures not very clear should dpi much more than 300.

8.Language can be more polish.

Comments on the Quality of English Language

minor revision

Author Response

Dear reviewer,

Thank you very much for taking the time to review our manuscript and provide crucial feedback. We have made revisions based on your suggestions, and the updated content is as follows:

1.This research focused on The Gastric Cancer Immune Prognostic Score (GCIPS) Showed Potential in Predicting an Unfavorable Prognosis for Gastric Cancer Patients who Undergoing Immune Checkpoint Inhibitors, after check the pubmed, there were  not so many articles aboult this topic, so this manuscript was very prospective and significant.

Response: Thank you for your recognition! it serves as a tremendous motivation for our work.

2.This manuscript foucus on clinical problems of gastric cancer, with strong clinical value and importantce,very interesting research, and also met the submission topic of this journal,the results was real and the conclusion was convincing, but some places can be more perfect.

Response: Thank you for your acknowledgment! we will make revisions according to your feedback.

  1. In MATERIALS AND METHODS part, GCIPS not carefully introduced should be in detail.To be fact, I donot understand what was GCIPS. “employed a random number table to divide the patients into a training set (n=201) and a validation set (n=101).” have some references?

Response: Thank you for your questions. Our study primarily focuses on the analysis of a comprehensive set of blood parameters before treatment to establish and validate GCIPS. The construction of GCIPS involves a stepwise selection process of blood parameters impacting patient prognosis. This phase constitutes a significant portion of our research efforts. To provide a clearer presentation of our study process, we have detailed the establishment and validation of GCIPS in the Results section. This enhances the overall coherence and logic of our manuscript, making it more reader-friendly and facilitating a better understanding of our approach. Therefore, we present a detailed overview of GCIPS in the Results section.

  1. Can this system GCIPS used for lung cancer or liver cancer?

Response: In our study, we have initially established and validated GCIPS in gastric cancer. Our results demonstrate significant potential for predicting the prognosis of gastric cancer patients undergoing ICIs. However, its prognostic value in other cancers requires further validation through subsequent research. We are eager to explore this in future studies, but at present, we cannot conclusively determine whether GCIPS can predict the prognosis of patients with other cancers.

  1. GCIPS was whole score, not only one index score, maybe better for Prognostic tracking, but how to Guiding treatment? Because during treatment, We may foucus on the main problem.

Response: Thank you for your questions. GCIPS is established based on patients' pre-treatment blood parameters, allowing it to reflect the patients' status before therapy. Clinical practitioners can use GCIPS to easily identify patients who may benefit from ICIs, and at the same time, stratify patients' risk through GCIPS to discover high-risk individuals for timely intervention. It holds certain potential applications in clinical settings. Relevant content has also been added to the discussion section of the manuscript.

6.I know now training and validation now was hot but all was retrospective study, can you use this system to apply in the clinical medicine really.

Response: ICIs rely on the patient's immune system to attack tumor cells, making them more dependent on the overall health status of the patient. Therefore, indicators reflecting the patient's inflammatory and nutritional status play a more crucial role in ICIs than in other treatments. The significant challenge faced by ICIs in gastric cancer at present is their lower response rate. Common biomarkers such as PD-1/PD-L1 expression, MSI, etc., cannot fully cover patients sensitive to ICIs. This underscores the increased significance of exploring new biomarkers. GCIPS, as an index established based on patients receiving ICIs, still holds substantial clinical potential in the current context.

7.Some figures not very clear should dpi much more than 300.

Response: Thank you for your feedback; we have adjusted all the images in the manuscript to 300dpi.

8.Language can be more polish.

Response: Thank you very much. We also found the language issues in our manuscript. Therefore, we have opted for the professional language editing service offered by your journal, enhancing the language quality of our manuscript.

Finally, allow me to express my gratitude once again for the work you have done on our manuscript. If there are any areas that need improvement, please do not hesitate to let us know, and we will make every effort to address them promptly.

Best wish to you!

Sincerely,

Hongjiang Song

Reviewer 3 Report

Comments and Suggestions for Authors

Dear authors,

After reviewing this study, with all due respect, this study is not on topic. The topic "The Gastric Cancer Immune Prognostic Score (GCIPS) Showed Potential in Predicting an Unfavorable Prognosis for Gastric Cancer Patients who Undergoing Immune Checkpoint Inhibitors" indicates the content of this study is highly related to the clinical outcome or clinical response of patients with gastric cancer treated with ICIs. However, since Materials and Methods section, description regarding to ICIs are missing until Discussion section. Moreover, the ICI regimens of enrolled subjects are all missed. In my opinion, this study is not developing prognostic factors for GC patients with ICI treatment, rather than prognostic factor for median/advanced GC patients.

Author Response

Dear reviewer,

Thank you for raising such critical questions. It is indeed a significant flaw that our study, focused on establishing biomarkers in immunotherapy patients, lacks specific descriptions of patients' immunotherapy. This study relies on data from various past clinical trials involving ICIs. Specifically, the study includes a total of 302 patients, all of whom underwent multiple cycles of ICI treatment. Among them, 203 patients participated in three clinical trials receiving camrelizumab (Clinical Trial Registration Numbers: CTR20200708, CTR20200045, CTR20190072), and 61 patients participated in another clinical trial receiving toripalimab (Clinical Trial Registration Number: CTR20212739). The remaining 38 patients, not enrolled in clinical trials, voluntarily opted for various ICIs, including toripalimab, pembrolizumab, camrelizumab, and sintilimab.

Finally, allow me to express my gratitude once again for taking the time to review our manuscript. Your feedback serves as a tremendous driving force for our improvement! If there are any areas that need further enhancement, please do not hesitate to inform us, and we will make every effort to address them promptly!

Best wish to you!

Sincerely,

Hongjiang Song

Reviewer 4 Report

Comments and Suggestions for Authors

The article by Yanjiao Zuo et al. titled " The Gastric Cancer Immune Prognostic Score (GCIPS) showed potential in predicting an unfavorable prognosis for gastric cancer patients who undergoing immune checkpoint inhibitors" offers valuable insights; however, there are several issues that require attention:

1. Abstract need to rewrite according to the author instructions

2. The introduction lacks sufficient background information about the current landscape of gastric cancer treatment and the role of immunotherapy. Providing a brief overview of the current treatment strategies and the challenges associated with identifying patients who can benefit from immunotherapy would strengthen the introduction.

3. The authors mention the value of classical inflammatory and nutritional markers in predicting prognosis but fail to provide specific examples or references to support this claim. It would be beneficial to include relevant studies or evidence to validate this statement.

4. The authors state that proposing a novel biomarker based on the latest treatment strategies is necessary, but they do not clearly explain why the treatment strategies for gastric cancer have undergone significant changes. Elaborating on these changes and their impact on prognosis prediction would enhance the argument for the need for a novel biomarker.

5. While the authors mention that GCIPS demonstrated excellent performance in survival analyses across all subgroups, they do not provide specific details or statistical measures to support this claim. Including survival curves or hazard ratios would provide a clearer understanding of the performance of GCIPS.

6. The authors state that GCIPS exhibited the highest prognostic value, surpassing even TNM stage and radical resection, but they do not provide comparative data or statistical analysis to support this claim. Including a comparison of the prognostic value of GCIPS with other established factors would strengthen their argument.

7. The authors briefly mention the validation set confirming the accuracy and stability of the results but do not provide any specific details or statistical analysis on this validation. Including validation statistics or a comparison of the performance between the test and validation sets would strengthen the study's findings.

8. Authors mention that the proposal of GCIPS provides a new reference for developing immunotherapy strategies, but they do not elaborate on how GCIPS can be practically applied or integrated into clinical practice. Discussing the potential implications and practical applications of GCIPS would enhance the relevance of the study.

Comments on the Quality of English Language

Moderate editing of English language required

Author Response

Dear reviewer,

Thank you very much for taking the time to work on our manuscript amid your busy schedule. Your suggestions are a tremendous driving force for our improvement. We have made modifications to the manuscript based on your feedback, and the revised content is as follows:

  1. Abstract need to rewrite according to the author instructions

Response: Thank you for your questions. We carefully reviewed the writing requirements for the abstract outlined in the author guidelines and made the necessary modifications. Additionally, we conducted language editing to enhance the overall linguistic quality of the manuscript. However, our study encompasses two primary aspects: the establishment of GCIPS and the investigation of its prognostic value. While we attempted to provide concise descriptions of both aspects in the abstract, it resulted in the word count exceeding 200 words. Regrettably, we found that simply condensing the descriptions did not allow us to meet the 200-word limit, and omitting descriptions of either aspect would hinder readers' understanding of our work. Therefore, we have temporarily refrained from reducing the word count. Nevertheless, we also have a plan, which involves removing the description of the GCIPS establishment process and streamlining other content. If you deem it necessary, we will make every effort to trim the abstract to 200 words while preserving its clarity.

  1. The introduction lacks sufficient background information about the current landscape of gastric cancer treatment and the role of immunotherapy. Providing a brief overview of the current treatment strategies and the challenges associated with identifying patients who can benefit from immunotherapy would strengthen the introduction.

Response: Thank you for your suggestion. We have added a description of the current status and challenges of immune checkpoint inhibitor (ICI) application in gastric cancer. While immunotherapy holds significant potential, it is not universally effective for all patients. In gastric cancer, only a small subset of patients demonstrates sensitivity to ICIs. Additionally, currently available biomarkers such as PD-1/PD-L1 expression and MSI not only incur high costs but also fail to comprehensively cover patients who may benefit from ICIs. This underscores the urgent need for a reliable biomarker to accurately predict patient responses and provide robust support for treatment strategies. The corresponding content has also been added to the manuscript.

  1. The authors mention the value of classical inflammatory and nutritional markers in predicting prognosis but fail to provide specific examples or references to support this claim. It would be beneficial to include relevant studies or evidence to validate this statement.

Response: Thank you for your question. In the introduction, we have introduced some studies on the application of classical inflammatory and nutritional markers in ICIs, which indeed enhances the persuasiveness of the manuscript.

  1. The authors state that proposing a novel biomarker based on the latest treatment strategies is necessary, but they do not clearly explain why the treatment strategies for gastric cancer have undergone significant changes. Elaborating on these changes and their impact on prognosis prediction would enhance the argument for the need for a novel biomarker.

Response: Thank you for your invaluable suggestion. We have added a description of the progress in gastric cancer treatment and highlighted the significant implications of this study in the field of immunotherapy for gastric cancer. With the advancement of medical technology, the treatment strategy for gastric cancer has evolved from singular approaches to a comprehensive combination of surgery, chemotherapy, targeted therapy, immunotherapy, and psychological treatment. This integrated treatment strategy has significantly prolonged the survival period for patients. In this context, the overall systemic condition of the patients becomes particularly crucial. The relevant content has also been added to the manuscript.

  1. While the authors mention that GCIPS demonstrated excellent performance in survival analyses across all subgroups, they do not provide specific details or statistical measures to support this claim. Including survival curves or hazard ratios would provide a clearer understanding of the performance of GCIPS.

Response: Thank you for your question. Initially, we were concerned that describing all survival curves would make the language of the manuscript cumbersome and significantly increase repetition rate. Therefore, considering that the relevant results have been displayed in the survival curve plots, after describing the main survival curves, we briefly summarized other subgroup analyses. However, this was not accurate enough. Therefore, we have added descriptions of the analysis values for all survival curves, primarily the chi-square values. This allows readers to understand the specific differences more accurately in survival curves.

  1. The authors state that GCIPS exhibited the highest prognostic value, surpassing even TNM stage and radical resection, but they do not provide comparative data or statistical analysis to support this claim. Including a comparison of the prognostic value of GCIPS with other established factors would strengthen their argument.

Response: The conclusion was derived from nomograms and HR values obtained through multivariate analysis. Initially, researchers incorporated factors significantly impacting patient prognosis into a nomogram to establish a predictive model for estimating patient survival probability. In the nomogram, different indicators receive varying maximum scores due to differences in their predictive values in the model. Specifically, the greater the impact on patient prognosis, the longer the corresponding indicator's "line" in the nomogram. Innovatively, we explored the prognostic value of the target indicator by constructing a nomogram, rather than using it to predict patient survival probability. In the nomograms we generated, including GCIPS, TNM staging, and curative resection, GCIPS had the longest "line," indicating the highest prognostic value. On the other hand, GCIPS also exhibited the highest HR value in the multivariate analysis. The HR value directly reflects the severity of the associated indicator, with values greater than 1 indicating higher risk and values less than 1 indicating lower risk. Through these two aspects, we can conclude that GCIPS has the highest predictive value, even surpassing TNM staging and curative resection.

  1. The authors briefly mention the validation set confirming the accuracy and stability of the results but do not provide any specific details or statistical analysis on this validation. Including validation statistics or a comparison of the performance between the test and validation sets would strengthen the study's findings.

Response: Thank you for your question. We provided more detailed descriptions of the validation set. In Table 1, we have listed all clinical and pathological information for the patients in the validation set. Additionally, we conducted a correlation analysis between the test set and the validation set, revealing no statistically significant differences in clinical and pathological data between the two datasets, forming the basis for all subsequent analyses. Subsequently, we plotted the time-ROC curve for GCIPS in the validation set and compared it with the time-ROC curve in the test set. The AUC in both datasets remained at an acceptable level, confirming the stability of GCIPS's high predictive value. Finally, the survival curves in the validation set further confirmed the high prognostic value of GCIPS.

  1. Authors mention that the proposal of GCIPS provides a new reference for developing immunotherapy strategies, but they do not elaborate on how GCIPS can be practically applied or integrated into clinical practice. Discussing the potential implications and practical applications of GCIPS would enhance the relevance of the study.

Response: Thank you for your suggestion. We have added a description of the clinical application of GCIPS. In this study, GCIPS, created based on pre-treatment blood indicators, demonstrated significant prognostic value in patients receiving ICIs. This provides clinicians with an easily accessible biomarker to assess patients who may benefit from ICIs. Additionally, GCIPS may assist clinicians in risk stratification, enabling the identification of high-risk patients for timely intervention. The potential value of GCIPS in clinical practice deserves further exploration. The corresponding content has also been added to the manuscript.

Finally, please allow me to express our gratitude once again for the work you've dedicated to reviewing our manuscript amidst your busy schedule. If there are any further areas that need improvement, please do not hesitate to inform us, and we will make every effort to revise accordingly.

Best wish to you!

Sincerely,

Hongjiang Song

Reviewer 5 Report

Comments and Suggestions for Authors

In this study, the authors established a gastric cancer immune prognostic model (termed GCIPS) through comprehensive blood parameter analysis before immunotherapy. They identified WBC, LYM and INR as independent prognostic factors for gastric cancer patients.  Importantly, GCIPS was associated with PFS and OS which is independent with radical resection and TNM stage. Furthermore, they conducted nomograms incorporating GCIPS and other pathological factors which exhibiting high predictive accuracy. The authors have done a lot of work to support their conclusions. Unfortunately, the design of the article has major flaws, and the language needs to be improved. Below are questions and suggestions:

1.       Which Immune checkpoint inhibitor was used in the gastric cancer patients in your study? Were patients taking the same inhibitor? If not, is GCIPS associated with the difference?

2.    Before ICIs treatment, the patients should take the detection of PD-1, PD-L1 or CTLA-4? How are the expression of these proteins? Is GCIPS associated with their expression?

3.    There are only 28 factors in blood parameters. Lasso regression analysis may not be appropriate for this study. Although some factors are neither statistically significant with survival nor independent prognostic factors, they may be essential for improving the accuracy of predictive models.

4.    In this study, ALB was excluded due to multicollinearity by Lasso regression analysis. From a clinical perspective, what is the role of ALB and whether it is important. Please explain it clearly in the article. And what is the reason for using 303 cycles of validation in Lasso regression analysis, not 300 or 1000?

5.    Please add the information of Lasso regression analysis in the description of Statistical Analysis.

6.    It is noted that your manuscript needs careful editing by someone with expertise in technical English editing paying particular attention to English grammar, spelling and sentence structure. e.g. “In this study, we established the independent blood parameters affecting patients in the test set through Cox regression analysis and Lasso regression analysis, identifying WBC, LYM, and INR.” It should be “identifying WBC, LYM, and INR as independent prognostic factors”.

7.    In the manuscript, some expressions are imprecise. e.g. “Additionally, GCIPS, TNM stage and radical resection were identified as independent prognostic factors in this study.” TNM stage and radical resection are independent prognostic factors for both PFS and OS.

Comments on the Quality of English Language

This manuscript needs careful editing by someone with expertise in technical English editing paying particular attention to English grammar, spelling and sentence structure.

Author Response

Dear reviewer,

Thank you for reviewing my manuscript amidst your busy schedules. Your suggestions are a significant driving force for our work. We have made revisions and provided explanations in accordance with your feedback. The updated content is as follows:

  1. Which Immune checkpoint inhibitor was used in the gastric cancer patients in your study? Were patients taking the same inhibitor? If not, is GCIPS associated with the difference?

Response: Thank you very much for your questions. The development of immune checkpoint inhibitors has been rapid, and currently, there are several drugs available. Our study indeed includes patients who have been treated with different immune checkpoint inhibitors. However, despite the diversity of immune checkpoint inhibitors, their overall mechanisms of action still revolve around enhancing the immune system's recognition and elimination of tumor cells by targeting PD-1, PD-L1, or CTLA-4 receptors. This immune system-dependent anti-tumor effect is closely linked to the overall condition of the patients. Therefore, as a biomarker reflecting the overall immune and inflammatory status of the patients, GCIPS is not greatly influenced by the specific differences in immune checkpoint inhibitors. A substantial body of previous research has also been conducted under these circumstances. Mezquita et al. established a novel Lung Immune Prognostic Index (LIPI) by collecting data from 446 non-small cell lung cancer patients undergoing Immune Checkpoint Inhibitor (ICI) treatment. Their patients received various ICIs, including nivolumab, pembrolizumab, atezolizumab, durvalumab, and durvalumab-ipilimumab. Ultimately, they revealed the significant prognostic value of LIPI, which has garnered widespread recognition [1]. Zheng et al. and Sun et al. conducted research on classical inflammatory and nutritional markers without specifically distinguishing the use of immune checkpoint inhibitors, and also meaningful results were obtained [2, 3].

  1. Before ICIs treatment, the patients should take the detection of PD-1, PD-L1 or CTLA-4? How are the expression of these proteins? Is GCIPS associated with their expression?

Response: Thank you for your inquiry. Although our patients were enrolled in multiple clinical trials, we were not specific participants in those trials. Consequently, the collection of patient data was conducted through the electronic medical record system, and we were unable to obtain specific PD-1, PD-L1 expression information for some patients. In this scenario, over half of the patients lacked PD-1, PD-L1, or CTLA-4 expression information. Analyzing results based on this situation would be inaccurate. Therefore, considering that it does not impact our main findings, we have removed the analysis related to PD-1 and PD-L1 expression. If you believe that this information is essential, we will not hesitate to include the relevant details.

  1. There are only 28 factors in blood parameters. Lasso regression analysis may not be appropriate for this study. Although some factors are neither statistically significant with survival nor independent prognostic factors, they may be essential for improving the accuracy of predictive models.

Response: Thank you for your question. In Cox survival analysis, it is common to first identify potential prognostic factors through univariate analysis and then perform multivariate analysis on these factors to improve the accuracy of the prognostic model. This approach has been widely applied in most prognosis studies. We also used this method in the subsequent analysis of patient clinical and pathological information. However, to maintain accuracy as much as possible, we initially included all blood parameters as continuous variables in the Cox survival analysis when establishing GCIPS. This approach led to a significant issue – multicollinearity. Blood parameters reflect the overall condition of patients, and there are many potential connections between them, with some factors even having calculated correlations (e.g., TP=ALB+GLOB). To address this problem, we introduced Lasso regression analysis between univariate and multivariate analysis. Lasso regression analysis is generally suitable for analyzing and verifying many factors, commonly used in the concentrated analysis of numerous genes. However, Lasso regression analysis also has a significant advantage, namely, it can discover potential multicollinearity among various factors and exclude it. This exclusion is achieved objectively in Lasso analysis. Specifically, if there is multicollinearity between two factors, Lasso will exclude the factor with a smaller impact on the prognostic model and retain the one with a larger impact. Therefore, the role of Lasso regression in this study is irreplaceable. Any potential multicollinearity could make the conclusions of this study unreliable.

  1. In this study, ALB was excluded due to multicollinearity by Lasso regression analysis. From a clinical perspective, what is the role of ALB and whether it is important. Please explain it clearly in the article. And what is the reason for using 303 cycles of validation in Lasso regression analysis, not 300 or 1000?

Response: As mentioned above, Lasso regression analysis objectively excludes indicators with multicollinearity through calculations. Therefore, the exclusion of ALB is objective. This does not mean that ALB has no impact on patients but rather signifies that its value in the prognostic model is not as significant as TP. Lasso regression analysis does not allow for manual control of the number of iterations but rather self-iterates until it reaches the optimal λ value and automatically stops.

  1. Please add the information of Lasso regression analysis in the description of Statistical Analysis.

Response: Lasso regression analysis is based on a complex algorithm, and its expertise goes beyond the scope of medicine. Many studies, including those involving Lasso, often report only the λ values. In our manuscript, we provided a description of the iteration process and the exclusion of indicators in the Lasso model, going beyond the routine reporting. However, if you feel that a detailed introduction to the entire algorithmic system is necessary, we will not hesitate to gather relevant content and add it to the manuscript.

  1. It is noted that your manuscript needs careful editing by someone with expertise in technical English editing paying particular attention to English grammar, spelling and sentence structure. e.g. “In this study, we established the independent blood parameters affecting patients in the test set through Cox regression analysis and Lasso regression analysis, identifying WBC, LYM, and INR.” It should be “identifying WBC, LYM, and INR as independent prognostic factors”.

Response: We have also found certain language issues in our manuscript. Therefore, we have opted for the professional language editing service provided by your esteemed journal, significantly enhancing the language quality of our manuscript.

  1. In the manuscript, some expressions are imprecise. e.g. “Additionally, GCIPS, TNM stage and radical resection were identified as independent prognostic factors in this study.” TNM stage and radical resection are independent prognostic factors for both PFS and OS.

Response: Thank you for your valuable suggestions. We have made corresponding modifications in the manuscript.

If there are any further areas that require improvement, please do not hesitate to let us know, and we will make every effort to enhance them.

Best wish to you!

Sincerely,

Hongjiang Song

Reference

  1. Mezquita, L, Auclin, E, Ferrara, R, et al. Association of the Lung Immune Prognostic Index With Immune Checkpoint Inhibitor Outcomes in Patients With Advanced Non-Small Cell Lung Cancer. JAMA ONCOL. 2018; 4 (3): 351-357. doi: 10.1001/jamaoncol.2017.4771
  2. Zheng, F, Meng, Q, Zhang, L, et al. Prognostic roles of hematological indicators for the efficacy and prognosis of immune checkpoint inhibitors in patients with advanced tumors: a retrospective cohort study. World J Surg Oncol. 2023; 21 (1): 198. doi: 10.1186/s12957-023-03077-8
  3. Sun, H, Chen, L, Huang, R, et al. Prognostic nutritional index for predicting the clinical outcomes of patients with gastric cancer who received immune checkpoint inhibitors. Front Nutr. 2022; 9 1038118. doi: 10.3389/fnut.2022.1038118

Reviewer 6 Report

Comments and Suggestions for Authors

The purpose of the research is to follow patients with stomach cancer who are undergoing immunotherapy. A key goal is the determination of positive immunotherapy. The authors followed up the study of classic inflammatory and nutritional markers for predicting patients undergoing immunotherapy.

According to the authors, it is necessary to propose a new biomarker based on the latest treatment strategies, so in this study, a gastric cancer immune prognostic index (GCIPS) was created by comprehensive analysis of blood parameters before immunotherapy, using regression analysis of Coke.

The content of white blood cells, lymphocytes and INR, GCIPS not only demonstrated excellent results in survival analyzes in all subgroups, but was also identified as an independent prognostic factor in this study. GCIPS shows the highest prognostic value, surpassing even TNM staging and radical resection. Analysis of the validation set further confirmed the accuracy and stability of the results. The GCIPS proposal provides a new guideline for developing immunotherapy strategies.

Results: The study was retrospective and the results used were well described and presented in eight tables and eight figures.

Discussion: I have no comments on the discussion.

Conclusion: this section can be expanded.

Comments on the Quality of English Language

 Extensive editing of English language required

Author Response

Dear reviewer,

Thank you very much for taking the time to review our manuscript. Your approval is a tremendous motivator for our progress! We have also acknowledged certain language issues in our manuscript. Therefore, we have opted for the professional language editing service provided by your esteemed journal, significantly enhancing the language quality of our manuscript. If there are any further areas that require improvement, please do not hesitate to let us know, and we will make every effort to enhance them.

Best wish to you!

Sincerely,

Hongjiang Song

Round 2

Reviewer 4 Report

Comments and Suggestions for Authors

Accept in present form

Comments on the Quality of English Language

Minor editing of English language required

Author Response

Thank you for your recognition and suggestions. We have significantly improved the quality of our manuscript by utilizing the excellent language editing services provided by your journal. Additionally, we have made every effort to review and revise the newly added content that has not undergone language editing.

Reviewer 5 Report

Comments and Suggestions for Authors

The authors provided detailed explanations and corrections. Please add the information of Lasso regression analysis in the description of Statistical Analysis.

  1.  

Author Response

Thank you for your suggestion. We have added a brief description of Lasso regression analysis in the "Statistical Analysis" section. Specifically as follows: "Additionally, we utilized Least Absolute Shrinkage and Selection Operator (Lasso) regression analysis to alleviate potential multicollinearity. Lasso regression analysis is a statistical method used for feature selection and regression analysis. It achieves sparsity of unimportant variables in the model by penalizing model parameters, effectively addressing multicollinearity. The key feature of Lasso regression is its ability to shrink the coefficients of predictive variables with minimal impact on the target variable to zero, thus facilitating feature selection. By adjusting the regularization parameters (λ value), Lasso regression can identify variables that significantly contribute to the predictive variables, enhancing the model's generalization ability and interpretability."